**Subject Area:**
neuroscience

*Drosophila*, neural circuits, hunger, feeding behaviour, food-seeking behaviour

**Author for correspondence:**
Suewei Lin
e-mail: sueweilin@gate.sinica.edu.tw

A contribution to the special collection commemorating the 90th anniversary of Academia Sinica.

# Neural basis of hunger-driven behaviour in *Drosophila*

Suewei Lin[1,2], Bhagyashree Senapati[1,2] and Chang-Hui Tsao[1]

[1]Institute of Molecular Biology, Academia Sinica, Taipei, Taiwan, Republic of China
[2]Molecular and Cell Biology, Taiwan International Graduate Program, Academia Sinica and Graduate Institute of Life Sciences, National Defense Medical Center, Taipei, Taiwan, Republic of China

(iD) SL, 0000-0001-7079-7818

Hunger is a motivational state that drives eating and food-seeking behaviour. In a psychological sense, hunger sets the goal that guides an animal in the pursuit of food. The biological basis underlying this purposive, goal-directed nature of hunger has been under intense investigation. With its rich behavioural repertoire and genetically tractable nervous system, the fruit fly *Drosophila melanogaster* has emerged as an excellent model system for studying the neural basis of hunger and hunger-driven behaviour. Here, we review our current understanding of how hunger is sensed, encoded and translated into foraging and feeding behaviours in the fruit fly.

## 1. Introduction

Hunger is an internal state elicited by lack of nutrients and energy in the body. It is difficult to establish if the subjective feeling of hunger is unique to human, but the foraging and consumptive behaviours evoked by hunger are nearly universal among mobile animals. Since foraging for food is costly with respect to energy and physical risk, hunger can be considered a guidance signal ensuring that animals only seek food when there is a need [1]. Defects in the hunger system can lead to malnutrition, obesity, eating disorders and even death. Hunger has very broad effects on an animal's behaviour. It can increase an animal's risk tolerance [2], elevate its locomotion [3–5], change its sensitivity to external stimuli [6,7] and affect its decision-making [8,9]. Effectively, hunger sets a goal—acquiring food—for an animal, and the animal can use whatever means are available to it to achieve that goal. Given these broad effects of hunger, unravelling its underlying neural mechanisms can be challenging. Compared to the extreme complexity of the mammalian brain, the nervous system of the fruit fly *Drosophila melanogaster* is simpler and genetic tools for labelling defined neuronal populations in the fly are more developed. These features have made the fly an excellent model for establishing fundamental neural principles of hunger. In this article, we review what has been learned from both fly larvae and adult flies in terms of how nutrient needs are sensed, how these needs are encoded as diverse hunger and satiety signals, and how these signals are translated by both peripheral and central neural circuits to elicit foraging and feeding behaviour. Since neural mechanisms in larvae and adult flies may not always be the same, we always clarify if a conclusion is based on larval or adult studies.

## 2. Sensing nutrient needs

Flies change their food preferences in response to lack of calories, amino acids or salts. This nutrient-specific hunger-driven behaviour implies the existence of internal sensors for specific food components, and several such molecular sensors have been identified in the fly. These sensors monitor nutrient levels and

royalsocietypublishing.org/journal/rsob    Open Biol. 9: 180259

regulate fly feeding behaviour accordingly when a nutrient concentration falls below a normal level.

Sugar is a major nutrient in food, and glucose is a primary source of energy in the body. Circulating sugar level is a good estimate for body energy state. Therefore, it is not surprising that the fly nervous system is equipped with the ability to directly sense glucose levels. Multiple lines of evidence show that flies can detect the nutrient value of sugar independently of its taste. Adult flies lacking both Gr5a and Gr64a, the taste receptors for sugar, still exhibit a preference for sucrose over plain agar in a two-choice assay after prolonged starvation [10]. This sugar preference appears to be correlated with haemolymph sugar content, and it can be enhanced by a drug that reduces circulating glucose and trehalose in the haemolymph. Flies are also capable of learning the nutritional value of sugar. Pairing sugar with odours can condition adult flies to form an appetitive olfactory memory that lasts for days. This memory becomes less robust if sweet but non-nutritious sugar is used. However, the robustness of the memory can be restored if the non-nutritious sugar is supplemented with a tasteless but nutritious substance [11,12].

Several tissues or cell types in the fly are considered to be glucose-sensitive. Insulin-producing cells (IPCs) in the brain and the corpora cardiaca that releases adipokinetic hormone (AKH) control feeding and foraging behaviours in both larvae and adult flies [4,13–16], and glucose contrastingly regulates their activities. Although direct evidence is still lacking, it is proposed that the glucose sensitivity of these cells is mediated by a mechanism similar to that of mammalian pancreatic α- and β-cells [17,18]. Fat body in the fly is a nutrient-sensing tissue equivalent to mammalian liver and adipose. Larval fat body has also been shown to sense glucose via a G protein-coupled receptor, Bride of sevenless (BOSS) [19]. BOSS protein contains a fragment homologous to a trehalose taste receptor, but it responds specifically to glucose, i.e. not to trehalose or sucrose. Furthermore, *boss* mutant adult flies and flies in which *boss* has been specifically knocked down in fat body exhibit increased food intake [20]. In addition to sensing glucose, gustatory receptor 43a (Gr43a) functions as a fructose receptor in the adult fly brain [21]. Gr43a is expressed in two to four neurons per hemisphere of the central brain of adult flies. These neurons respond to fructose, but not other tested sugars (including sucrose, glucose and trehalose). This fructose response is Gr43a-dependent and is completely abolished in *gr43a* mutant flies. Although glucose and trehalose are the main haemolymph sugars in insects, large increases in fructose concentration in the haemolymph have been observed after flies have eaten a sugar meal, even if the sugar consumed is glucose or sorbitol. Thus, haemolymph fructose can serve as an estimate for carbohydrate consumption. Consistent with their role as internal nutrient sensors, brain Gr43a neurons promote sugar intake in hungry flies and suppress sugar feeding in satiated flies [21].

Amino acids are another primary food nutrient. When they encounter food lacking essential amino acids, *Drosophila* larvae initially consume it as normal food but, after approximately 1 h, food intake is reduced by 20–25% compared with larvae that eat normal food. This delayed reduction in food intake is mediated by a small group of amino acid-sensitive dopaminergic neurons in the brain [22]. These neurons are activated when larval brain explant is perfused with an imbalanced amino acid mix, and this response requires the amino acid transporter *slimfast* and the intracellular amino acid-sensor GCN2. In

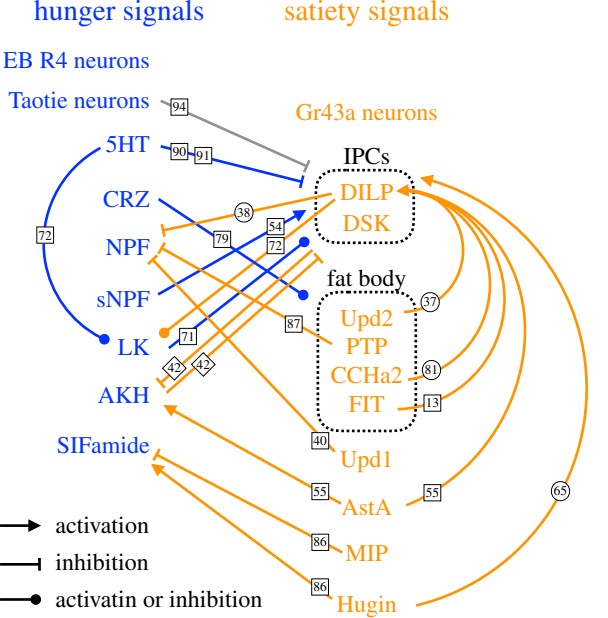

**Figure 1.** Hunger and satiety signals and their interactions. The numbers on the lines indicate the references in which evidence for the indicated interaction is presented. The shapes that outline the numbers denote whether the evidence supporting the indicated interaction are from larval studies (circle), adult studies (square) or both (diamond). Gr43a neurons, Taotie neurons and EB R4 neurons were identified in adult flies.

addition to neurons, larval fat body has also been demonstrated to use *slimfast* and TOR/S6 K to sense amino acids and secrete endocrine signals to regulate functions of the nervous system [23,24]. Adult flies also exhibit nutrient-specific hunger, exhibiting a preference for protein-rich food when they are deprived of amino acids [25,26]. Adult fat body and some neurons in the adult brain, including a group of dopaminergic neurons, have been shown to respond to protein starvation, but their sensory mechanisms remain unclear [13,27,28]. Moreover, the amino acid-sensitive TOR/S6 K pathway also appears to regulate amino acid state-dependent food preferences in adult flies, but the neurons on which TOR/S6 K signalling acts have not yet been identified [25,26].

Food components are complex. Our understanding of how the nervous system detects the need for specific nutrients remains profoundly insufficient. Whether there are additional internal sensors in the fly for other nutrients such as salt, lipids or minerals, as well as the molecular nature of these sensors, are exciting questions for future research.

# 3. Neural coding of hunger and satiety states

A diverse array of neuronal signals induced by hunger and satiety states has been identified in the fly (figure 1). Combinations of these signals can be considered representations of hunger states. Most of these signals are neuropeptides, which are modulatory and can potentially work long-range to control multiple neural circuits in the nervous system.

Insulin-like peptides (DILPs) and Unpaired 2 (Upd2) are the fly equivalents of mammalian insulin and leptin. The fly has eight different DILPs. Expression of the eight DILPs is tissue-specific and dynamic during development [29–31]. There is only one known insulin receptor in the fly, but insulin

signalling has been shown to regulate diverse biological processes, including cell growth, longevity, feeding and food foraging behaviour [6,15,29,32–34]. DILP2, 3 and 5 are co-expressed in adult and larval IPCs [35,36]. Secretion of DILPs from the IPCs is hunger-dependent [24,37]. When larvae are nutrient-deprived, DILPs accumulate in the IPCs, and their concentrations decline in the haemolymph. DILPs function as satiety signals that attenuate feeding motivation. Overexpression of DILPs in larval IPCs reduces feeding activity in starved larvae. By contrast, downregulation of DILP release promotes feeding in larvae and increases their acceptance of low-quality foods, suggesting elevated feeding motivation [38]. In adult flies, DILPs have also been shown to act as satiety signals modulating neural circuits that control feeding and food-seeking behaviour [6,16,32,39]. The nutrient state-dependent control of DILP release in larvae requires Upd2, which is secreted from fat body [37]. Under starvation conditions, upd2 transcript levels are greatly reduced. In the fed state, Upd2 acts on its receptor Domeless to activate the JAK/STAT signalling pathway in a population of GABAergic neurons that project onto the larval IPCs. Active JAK/STAT signal lowers the neuronal activity of these GABAergic neurons and releases their inhibitory effect on the IPCs, leading to DILP release. Human leptin fully rescues upd2 mutant fly phenotypes, suggesting that they are functional homologues. Intriguingly, despite the direct stimulatory effect of Upd2 on DILP release, knockdown of upd2 in fat body mainly impairs body growth and has a minimal effect on feeding and food-seeking behaviour in both larvae and adult flies [37,40]. Recently, Unpaired 1 (Upd1), another fly leptin-like peptide, was revealed to fulfil these roles of Upd2 upon upd2 knockdown in the adult fly [40]. Unlike Upd2, which is secreted from fat body, Upd1 is derived from a small cluster of neurons in the brain. Fed flies have higher levels of upd1 mRNA and less detectable proteins in Upd1-expressing neurons compared to hungry flies, suggesting that the nutrient-rich state may increase both transcription and release of Upd1. Importantly, pan-neuronal knockdown of upd1 in fed flies increases food intake and heightens their responses to food odour. Thus, Upd1 and Upd2 might work in concert to signal satiety states in the fly.

AKH is the fly analogue of mammalian glucagon. Akh is expressed in the corpora cardiaca from late embryo to adult stages [4]. Starvation induces AKH release into haemolymph to signal hunger [18]. AKH induces utilization of stored energy by stimulating lipolysis, glycogenolysis and trehalose release in larval and adult fat bodies [18,41]. AKH and DILP signals are mutually inhibitory. Ablation of IPCs increases AKH expression, whereas ablation of the corpora cardiaca enhances DILP3 expression in both larvae and adult flies [42]. Adult flies lacking AKH are more resistant to starvation and do not exhibit starvation-induced hyperactivity [4], phenotypes regulated by AKH via a small group of octopaminergic neurons in the brain [33]. AKH also regulates adult feeding behaviour by directly activating four interoceptive SEZ neurons (ISNs) in the suboesophegeal zone (SEZ), as well as by reducing bitter sensitivity in adult flies [16,43].

Another hunger signal in the fly, Neuropeptide F (NPF), is a homologue of mammalian Neuropeptide Y (NPY) [44]. NPF is expressed in the brain and endocrine cells of the midgut in both larvae and adult flies. The role of midgut NPF is less clear, but brain-derived NPF appears to facilitate feeding and foraging behaviour in starved flies. In larvae, NPF is expressed in four neurons in the brain. Ablation of these neurons results in reduced feeding behaviour, whereas broad overexpression of NPF in the nervous system prolongs the feeding phase of third instar larvae [45]. Furthermore, food-deprived larvae exhibit a higher tolerance for feeding on low-quality or noxious foods. Blocking neurotransmission of NPF-expressing neurons impairs this starvation-induced tolerance [15,38]. The adult brain contains approximately 30 NPF-positive neurons [46]. Activation of these NPF neurons increases gustatory sensitivity of adult flies to sugar and promotes food intake [43,47]. The adult NPF neurons express Domeless and are inhibited by the satiety signal Upd1 [40]. They also respond to olfactory inputs, and these responses are positively correlated with the attractiveness of the presented odour [48]. Therefore, NPF neurons may integrate sensory and internal state information to regulate foraging behaviour. NPF also mediates hunger-dependent expression of food memory. Starved adult flies can be taught to associate sugar with odours that have no innate appetitive value, and the resulting olfactory memory is only expressed when flies are hungry [49]. Stimulation of NPF neurons is sufficient to mimic the hunger state and promote sugar-odour memory expression in food-satiated adult flies. Consistently, npfr mutant adult flies fail to express this memory when they are food-deprived, behaving like non-hungry flies [50].

Flies have another NPY-like peptide, short neuropeptide F (sNPF). NPF and sNPF are not evolutionarily closely related [51]. Nevertheless, sNPF also regulates multiple aspects of hunger-driven behaviour. Overexpression or RNAi knockdown of snpf pan-neuronally promotes or suppresses feeding in adult flies, respectively [52]. sNPF regulates adult feeding behaviour partly by suppressing the synaptic release of bitter-sensing gustatory receptor neurons [43]. Furthermore, starvation increases sNPF signalling in odorant receptor neurons (ORNs) to elevate their sensitivity to food odours, thereby promoting foraging behaviour in adult flies [6]. sNPF has also been demonstrated to directly stimulate the expression of DILPs in larval and adult IPCs [53,54]. This scenario seems to contradict the proposed role of sNPF as a hunger signal. However, upregulation of DILPs by sNPF might relate more to regulating metabolism and growth, rather than feeding motivation. Given that sNPF is broadly expressed in the fly nervous system, it is also possible that sNPF functions as a co-transmitter in most neural circuits, including those involved in hunger-driven behaviours [51].

Allatostatin A (AstA) is another satiety signal that inhibits adult feeding behaviour when flies are fed [47,55,56]. Its receptors DAR-1 and DAR-2 are homologues of the mammalian galanin receptors [57,58]. Activation of AstA-expressing neurons reduces the proboscis extension reflex (PER) of hungry flies to sucrose as well as their food intake [47]. The same manipulation has no effect on PER in fed flies, suggesting that the AstA signal might already be maximized when flies are well fed. Consistent with the notion that AstA signal is high under fed conditions, flies on a low-nutrient diet (1% sucrose) exhibit downregulated expression of AstA and its receptor dar-2 in IPCs and AKH-producing cells in the corpora cardiaca [55]. Giving flies a high sugar diet (cornmeal) after nutrient restriction strongly increases the expression of AstA and dar-2. However, when a high-protein diet (yeast) is given, AstA expression is only modestly increased and dar-2 expression is unchanged. Interestingly, flies fed ad libitum normally prefer sucrose over yeast food, but when AstA neuronal activity is genetically increased in these flies, they shift their preference

royalsocietypublishing.org/journal/rsob    Open Biol. 9: 180259

royalsocietypublishing.org/journal/rsob    Open Biol. 9: 180259

to yeast food, suggesting that AstA might be a nutrient-specific satiety signal for sugar [55].

Myoinhibitory peptides (MIPs) have also been identified as a potent satiety signal in the adult fly brain [59]. The *mip* gene encodes five MIPs (MIP1-5) that are closely related [60]. Some MIPs have N-terminal sequences similar to mammalian Galanin [61]. Silencing MIP-expressing neurons makes food-satiated flies behave like hungry flies, showing elevated taste sensitivity to sugar, heightened olfactory sensitivity to food odours and increased food intake [59]. Similar phenotypes are also observed in *mip* mutant flies. By contrast, activation of MIP neurons in starved flies reduces their food intake, body weight and sensory sensitivity toward food. MIPs are expressed in about 70 neurons in the central nervous system, but it is not clear which neurons mediate the satiety effect or how MIP neurons sense the satiety state.

Hugin is a fly homologue of mammalian neuropeptide U [62]. It is expressed in 20 neurons in the larval SEZ with axonal projections to the ventral nerve cord, the pharynx, the protocerebrum, as well as the ring gland (the master larval neuroendocrine organ), where AKH-producing cells reside [27]. The connectivity patterns of hugin neurons in the adult brain are less well characterized, but are suggested to be similar to those of the larval brain [27]. Transcription of the *hugin* gene is affected by diet. Levels of *hugin* mRNA are decreased in both food-deprived larvae and larvae grown on a sugar-rich but amino acid-deficient diet, indicating that hugin may function as an amino acid-specific satiety signal. High *hugin* mRNA levels in larvae are correlated with low food intake and reduced food-seeking behaviour [27]. Furthermore, activation of hugin neurons reduces yeast food intake, whereas blocking them promotes feeding in both larvae and adult flies [27,63]. A subset of hugin neurons (hugin PC neurons) in the larval brain that project to the protocerebrum have also been shown to be part of the bitter gustatory pathway, linking bitter sensation to median neurosecretory cells, including IPCs [64,65]. Therefore, hugin neurons might represent a hub in the fly brain for integration of taste and internal nutrient level signals, facilitating the regulation of feeding motivation.

Leucokinin (LK) is an insect neuropeptide that was initially identified as regulating body water homeostasis [66]. It has since also been shown to control feeding behaviour and metabolism [67]. There is no known mammalian counterpart for LK, but its receptor (LKR) is homologous to the vertebrate tachykinin receptor [68]. LK is expressed in three sets of neurons in the adult fly nervous system: a pair of LHLK neurons in the lateral horn (LH), a pair of SELK neurons in the SEZ, and 11–12 pairs of ABLK neurons in the abdominal ganglia (AB) [69,70]. During feeding, *lk* and *lkr* mutant adult flies consume larger meals at a lower frequency than control flies [67]. These mutant flies also exhibit reduced food intake in long-term but not short-term feeding assays [71]. However, conditional activation or silencing of LK neurons in adult flies both result in decreased food intake [72], so the neural mechanisms underlying the contribution of LK to feeding behaviour are not fully understood. Apart from its impact on feeding behaviour, the LK pathway has been shown to mediate postprandial sleep in adult flies [73]. Flies sleep more after eating, and this postprandial sleep is mainly induced by protein and salt but not sucrose consumption. Activation of LK neurons reduces postprandial sleep, whereas *lk* knockdown has the opposite effect. These results suggest that LK may function as a nutrient-specific hunger signal. Furthermore, adult ABLK neurons express insulin receptor (InR) and the 5HT1B receptor for the hunger signal serotonin (see below) [72,74], and adult IPCs express LKR [71]. Knockdown of InR in adult ABLK neurons elevates LK levels in the cell bodies, but whether this is due to decreased LK release or increased LK production has not been determined. By contrast, knockdown of 5HT1B reduces LK levels, but again it is not clear whether the reduced LK level is due to increased LK release or reduced LK production. Although high-concentration serotonin treatment (10 mM) has been shown to silence ABLK neurons, this outcome could be an indirect consequence of strong neuronal bursting [72]. Notably, the 5HT1B receptor is not expressed in adult LHLK or SELK neurons, suggesting that the LK neurons in the brain and ventral ganglia are differentially regulated and may have different functions [71,72,75]. Complex results have also been observed when LKR is knocked down in adult IPCs, with LK signal seeming to negatively regulate DILP2 expression, but positively regulate DILP5 expression and DILP3 release [71]. Taken together, LK is likely to be part of the hunger regulatory network, but its detailed role remains to be resolved.

Corazonin (CRZ) is evolutionarily related to AKH [76]. In both larval and adult brains, CRZ is predominantly expressed in bilateral clusters of dorsolateral neurons (DLPs), which also express sNPF [77,78]. Chronic activation of CRZ neurons leads to increased food intake in adult flies [47], whereas knockdown of *crz* in these neurons has the opposite effect [79]. The CRZ receptor (CrzR) is expressed in adult salivary glands and fat body, and knockdown of *CrzR* in these peripheral tissues results in decreased food intake. However, knockdown of *CrzR* also has complex effects on the expression of multiple genes involved in regulating metabolism and stress [79]. Consequently, further studies are required to reveal the precise role of CRZ on hunger-driven behaviour.

Drosulfakinin (DSK) is the fly homologue of mammalian cholecystokinin. In addition to being expressed in several neuron groups in the fly brain, DSK is co-expressed with DILPs in a subset of larval and adult IPCs [80]. DSK appears to function as a satiety signal together with DILPs. RNAi knockdown of *dsk* specifically in these IPCs increases food intake in both larvae and adult flies, even when the proffered food is bitter or otherwise less palatable. A similar phenotype has been observed when these IPCs or all DSK-expressing cells are silenced. Furthermore, downregulation of *dsk* expression in adult IPCs leads to increased levels of *dilp2*, *3* and *5* transcript, suggesting compensatory regulation among these satiety-mediating neuropeptides [80].

A peptide hormone named female-specific independent of transformer (FIT) was recently discovered to be a protein-specific satiety signal in adult flies [13]. Protein but not sucrose or lipid consumption increases the expression of FIT. The protein-induced satiety effect on feeding is impaired in flies lacking FIT, whereas FIT overexpression reduces the preference of flies for protein food as well as their overall food intake after starvation. FIT is expressed in fat cells within the head of adult flies but not in the brain. However, it has been demonstrated that FIT secreted from fat cells executes its satiety effect by promoting DILP release from IPCs. FIT has not been detected in larvae, suggesting that it is an adult-specific satiety signal.

Another neuropeptide proposed to mediate hunger and satiety control is CCHamide-2 (CCHa2), an insect neuropeptide without a known counterpart in mammals. However, conflicting results have been obtained regarding the role of CCHa2 in feeding control. In one study, CCHa2 was found

royalsocietypublishing.org/journal/rsob    Open Biol. 9: 180259

to be expressed in larval fat body and in the midgut [81]. In that study, CCHa2 expression levels decreased in starved larvae and levels could be recovered by feeding yeast food or glucose to the larvae. Moreover, perfusion of CCHa2 onto larval brain explants can directly activate IPCs, in which the CCHa2 receptor (CCHa2-R) is enriched. Also, *ccha2-r* mutant larvae exhibit normal feeding activity, but their feeding phase during the third instar stage is prolonged. These results suggest that CCHa2 is a nutrient-stimulated satiety signal similar to Upd2 [37]. However, in other studies, CCHa2 has only been detected in the midgut but not fat body in both larvae and adult flies [82,83]. Furthermore, both larvae and adult flies lacking CCHa2 show reduced feeding activity, suggesting that CCHa2 is a hormone that stimulates feeding [83]. These conflicting results may partly be due to the pleiotropic functions of CCHa2 pathways, so more research is required to further elucidate the roles of CCHa2 in feeding regulation.

SIFamide (SIFa) has been shown as a hunger signal in adult flies. The receptor of SIFamide (SIFaR) is homologous to vertebrate gonadotropin inhibitory hormone receptor (GnIHR), but the sequences of SIFa and GnIH are not closely related [84,85]. SIFa is expressed in four neurons in the fly brain and their neurites innervate broadly in the brain, particularly densely in the antennal lobe, central complex and SEZ [86]. SIFa neuronal activity is elevated in starved flies. Acute activation of SIFa neurons alone is sufficient to increase the response of fed flies to food odour and to enhance their food intake, making them behave like hungry flies. However, although knockdown of SIFa completely abolishes starvation-induced sensitization of olfactory projection neurons to food odour, silencing SIFa neurons does not affect food intake in starved flies, suggesting that hunger may elicit recruitment of compensatory pathways to promote feeding. SIFa neurons are inhibited by satiety-encoding MIP neurons. Intriguingly, they are also positively regulated by another satiety signal, hugin [86]. The biological significance of hugin regulation in this scenario is not clear, but it highlights the complexity of hunger regulation.

Apart from neuropeptides, 6-pyruvoyltetrahydropterin (PTP) and serotonin are known to be satiety and hunger signals, respectively, in adult flies. PTP represents a curious case illustrating how peripheral nutrient-sensing tissue can communicate with central neural circuits that control feeding behaviour [87]. PTP is an intermediate in the biosynthesis pathway that produces the enzymatic cofactor tetrahydrobiopterin (BH4). There are three enzymes in this pathway—Purple (Pr), Punch (Pu) and Sepiapterin reductase (Sptr). Knockdown of *pr* or *pu* in fat body increases food intake, which can be rescued by feeding flies BH4. Intriguingly, knockdown of *sptr* in fat body has no effect on feeding behaviour. Instead, increased food intake has been observed when *sptr* is knocked down in NPF neurons. BH4 appears to inhibit NPF release through an unknown mechanism. Furthermore, expression of *pr* and *pu* in fat body is diet-dependent. Flies on a low-nutrient diet exhibit reduced *pr* and *pu* expression. Therefore, when flies are well fed, increased levels of Pr and Pu have been proposed to elevate synthesis of PTP, which is released from fat body, circulated to the brain and taken up by NPF neurons. In the NPF neurons, PTP is converted by Sptr to BH4, which inhibits the release of NPF that promotes feeding behaviour.

In contrast to the satiety function of BH4, activation of a small subset of serotoninergic neurons mimics starvation and induces a potent hunger response in food-satiated adult flies [88]. There are 25 of these hunger-inducing serotoninergic neurons per adult fly brain hemisphere and they project their neurites into broad areas. Artificial activation of these serotoninergic neurons in fed flies causes the flies to ingest the same amount of food as flies starved for 24 h. They also exhibit the same intensity of PER in response to sucrose solution and the same level of preference for nutritious sugar over sweet-only sugar as starved flies. However, serotoninergic neuronal activation does not mimic starvation-induced hyperactivity, which is mediated by octopamine [89], highlighting the modular nature of the hunger control mechanism. Serotonin may promote hunger-driven behaviours partly by inhibiting DILP expression in IPCs. IPCs express the serotonin receptor 5HT1A, and knockdown of 5HT1A in IPCs is reported to increase the expression of DILP2 and DILP5 [90,91].

Taotie neurons and a group of ellipsoid body (EB) R4 neurons in the adult fly brain have also been suggested to encode hunger [92–94]. The neurotransmitters/modulators released by these neurons are currently unknown, but Taotie neurons are likely be peptidergic [94]. Taotie neurons form a small cluster in the *pars intercerebralis* (PI) that also harbours neurosecretory cells including IPCs. Taotie neurons do not overlap with these IPCs, and their activation induces persistence of hunger signal. Fed flies with activated Taotie neurons consume the same amount of food as flies starved for at least 12 h. These flies also prefer nutritious sugar and yeast (i.e. just like the starved flies). Activation of Taotie neurons also reduces the release of DILPs from IPCs, suggesting that Taotie neurons promote hunger-driven behaviours partly through inhibiting the satiety signal of DILPs [94]. EB R4 neurons express *sodium/solute co-transporter-like 5A11* (*SLC5A11*) [93]. When flies are starved, transcription of *SLC5A11* increases in the R4 neurons, leading to enhanced neuronal excitability by inhibiting the *Drosophila* KCNQ potassium channel [92]. Overexpression of *SLC5A11* or activation of R4 neurons is sufficient to drive feeding behaviour in food-satiated flies, whereas silencing these neurons has the opposite effect. However, how starvation regulates *SLC5A11* expression in R4 neurons remains to be established.

The large number of neuromodulators involved in hunger-driven behaviour highlights the complexity of this motivational system, even in small insects. These neuromodulators do not work alone. Instead, they interact with each other to generate coordinated outputs (figure 1). Many of these modulators converge on IPCs, regulating their release of DILPs and presumably also DSK. Remote control of IPCs by fat body, as well as mutual inhibition between hunger- and satiety-mediating neuromodulators, are also prominent features of the control of this motivational state. Systems approaches that allow visualization and manipulation of multiple neuromodulators simultaneously will be needed to understand the dynamics of this intricate regulatory network. Furthermore, developing an understanding of the links between nutrient sensors and this regulatory network remains an important avenue for future research. In the following sections, we discuss how these hunger and satiety neuromodulators act on both central and peripheral neural circuits to control feeding and foraging behaviours.

# 4. Hunger-based modulation of feeding circuits

Consumption of appropriate nutrients to fulfil body requirements is the ultimate goal of hunger motivation. Starvation

sharpens the sensitivity of flies to food components, increases their preference for nutritious food and dampens their response to bitter compounds. Recent studies have begun to unravel the neural mechanisms underpinning these hunger modulations.

## 4.1. Hunger sensitizes sugar taste

Flies detect food components via a repertoire of taste-specific gustatory receptor neurons (GRNs). Sugar-sensing GRNs expressing gustatory receptor 5a (Gr5a) in adult flies show enhanced responses to sucrose upon starvation (figure 2a). The enhancement requires presence of the dopamine receptor DopEcR in the Gr5a neurons, and a single dopaminergic neuron (TH-VUM) is likely to be the source of dopamine in this case [43,95,96]. TH-VUM neurites broadly innervate the SEZ, also where Gr5a neurons innervate. The neuronal activity of TH-VUM is positively correlated with the starvation duration. Silencing TH-VUM decreases PER of starved flies to sucrose, whereas increasing the excitability of TH-VUM elevates PER to sugar in both fed and starved flies. This modulation of sugar sensitivity is likely initiated by the hunger signal NPF [43]. Activation of NPF neurons also increases the sugar response of Gr5a neurons. However, this enhanced effect disappears in flies with hypomorphic mutation of *DopEcR*. The link between NPF neurons and TH-VUM or Gr5a neurons is unclear. Knockdown of *npfr* in the TH-VUM neuron does not affect sugar sensitivity, suggesting that the effect of NPF on the TH-VUM-to-Gr5a GRN pathway is indirect.

## 4.2. Hunger desensitizes bitter taste

A parallel neural pathway regulates starvation-induced dampening of the response to bitter foods (figure 2a). Hungry larvae and adult flies show a higher tolerance to food containing bitter substances [15,43], and in adult flies this increase in bitter tolerance correlates with the decreased bitter response of Gr66a-positive bitter-sensing GRNs [43]. This hunger-dependent modulation of Gr66a neurons requires sNPF, but not NPF. Silencing a subset of sNPF neurons called lateral neurosecretory cells (LNCs) increases bitter sensitivity in hungry flies. LNCs project their axons to the SEZ, which is also innervated by Gr66a neurons. However, the effect of sNPF is likely to be indirect, acting through as yet unidentified GABAergic neurons. Furthermore, activation of hunger-mediating AKH neurons decreases bitter sensitivity in an sNPF-dependent manner, suggesting that AKH neurons may function upstream of or parallel to the sNPF pathway [43]. Two pairs of SEZ-innervating OA-VL neurons have also been shown to regulate starvation-induced bitter insensitivity in adult flies [97]. These OA-VL neurons release both octopamine and tyramine to directly potentiate Gr66a neurons in fed flies. The neural pathways upstream of the OA-VL neurons remain to be determined, but given that starvation strongly decreases their neuronal activity, sNPF-regulated GABAergic neurons are likely candidates.

## 4.3. Hunger modulates other neurons involved in sugar feeding

In addition to the neurons in the taste circuits, several other neurons have been identified as regulating sugar feeding. A pair of Fdg neurons in adult flies appear to act as command

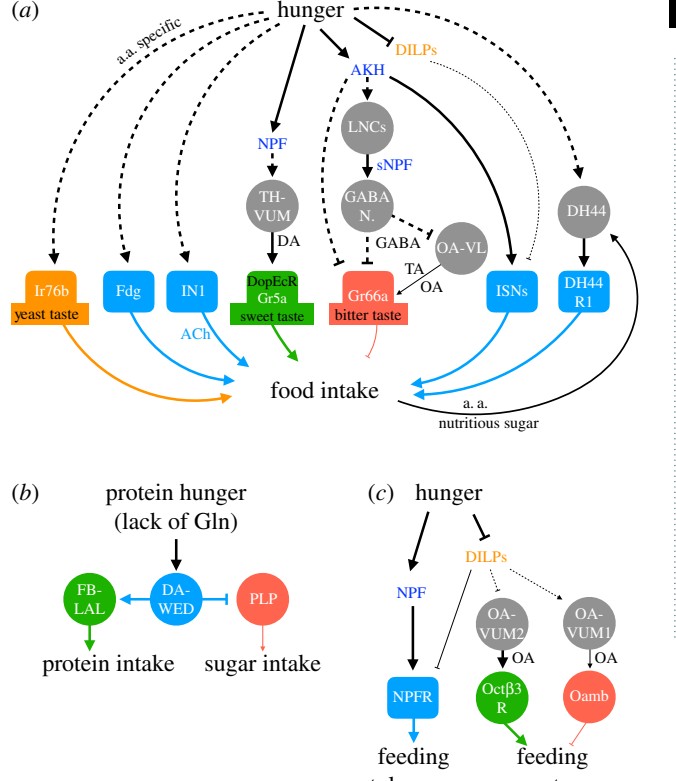

**Figure 2.** Hunger-based control of feeding circuits. (a) In adult flies, hunger modulates GRNs (orange, green and red) and SEZ neurons (light blue) to promote food intake. Starvation increases the release of NPF, which indirectly activates the dopaminergic TH-VUM neurons that in turn potentiate sweet taste-responsive Gr5a neurons via the dopamine receptor DopEcR. Starvation also increases the release of AKH, which indirectly activates sNPF-releasing LNCs. sNPF then activates as yet unknown GABAergic neurons that inhibit the bitter taste-responsive Gr66a neurons. The same GABAergic neurons may also inhibit OA-VL neurons that can potentiate Gr66a neurons by releasing tyramine (TA) and octopamine (OA). In addition, starvation potentiates yeast taste-responsive Ir76 neurons, Fdg command neurons, cholinergic IN1 neurons, and AKHR-expressing ISNs to promote food intake. Starvation also positively regulates DH44 neurons that promote feeding (via DH44R1-expressing neurons) in response to post-ingestive amino acid and nutritious sugar signals. Dashed lines indicate the regulation is indirect or its underlying mechanism is not fully understood. a.a., amino acids. (b) In the adult central brain, protein starvation (and particularly lack of glutamine) activates dopaminergic DA-WED neurons, which activate FB-LAL neurons to promote protein intake while inhibiting PLP neurons that promote sugar consumption. (c) In fly larvae, hunger increases feeding tolerance through an NPF pathway, while also enhancing feeding rate via OA neurons. Two populations of OA neurons, OA-VUM1 and OA-VUM2, act, respectively, through OAMB- and Octβ3R-expressing neurons to contrastingly regulate feeding rate. Additionally, DILPs (release of which is inhibited by hunger) also negatively regulate feeding tolerance and feeding rate. DILPs regulate feeding tolerance via NPFR neurons, but whether they regulate feeding rate through the OA neurons remains to be determined.

neurons, activation of which is sufficient to generate the full spectrum of the feeding motor programme, from proboscis extension to pharyngeal pumping and proboscis retraction (figure 2a) [98]. Fdg neurons are activated by sugar taste, but only when a fly is starved. Probably downstream of the Fdg neurons are several motor neurons in the SEZ that innervate muscles involved in feeding behaviour. Activating these neurons individually triggers a subset of the feeding motor

royalsocietypublishing.org/journal/rsob    Open Biol. 9: 180259

programme [99–101]. The upstream mechanisms relaying taste and internal state information to Fdg neurons are not clear. Twelve cholinergic interneurons (IN1) in the SEZ form synapses with sugar-sensing neurons (figure 2a) [102], but their relationship with Fdg neurons has not been established. IN1 activity is also modulated by hunger state. However, unlike Fdg neurons that respond to sugar taste, IN1 is activated by sucrose ingestion. Also, activation of IN1 does not directly trigger feeding behaviour, but it does increase the probability of sugar ingestion when a drop of sucrose solution is presented in close proximity to a fly. Four interoceptive SEZ neurons (ISNs) have also been identified as controlling sugar feeding in adult flies (figure 2a) [16], but whether ISNs connect to Fdg neurons remains undetermined. Activation of ISNs induces sugar consumption even in well-fed flies. However, interestingly, ISNs do not respond to sugar taste but encode information about hunger state. The activity of ISNs increases significantly in starved flies, and this modulation requires AKHR expressed in ISNs. Furthermore, application of hunger-promoting AKH to explant brains activates ISNs. Apart from AKH-mediated regulation, ISNs are indirectly inhibited by satiety-inducing DILPs, demonstrating a push and pull mechanism for satiety-state control. Intriguingly, ISNs are repressed by high haemolymph osmolarity caused by desiccation and they negatively regulate water consumption [16]. Thus, ISNs represent a signal convergence node in the fly brain where hunger and thirst motivations compete for behavioural expression.

## 4.4. Protein hunger modulates fly responses to amino acids

Like the sugar-sensing GRNs, gustatory neurons for yeast taste are regulated by internal amino acid levels (figure 2a) [103]. Ir76b-expressing GRNs in the proboscis of adult flies respond to yeast taste and are required for yeast intake. Amino acid-deprivation significantly increases the response of these GRNs to yeast. Amino acid-deprivation has no effect on the response of Gr5a neurons to sucrose, highlighting the nutrient specificity of hunger control. How amino acid hunger regulates these yeast-responsive GRNs remains to be identified. It would be interesting to investigate if protein-specific satiety signals, such as FIT and hugin, are involved in this regulation.

Amino acid starvation also promotes yeast feeding by regulating central brain circuits. Two dopaminergic neurons (DA-WED) in each hemisphere of the adult brain and that innervate the vedge neuropil are proposed to encode protein hunger (figure 2b) [28]. Silencing these DA-WED neurons decreases yeast intake but increases sucrose consumption, whereas activating these neurons enhances yeast intake but reduces sucrose consumption. Therefore, like overall hunger and thirst, nutrient-specific hunger motivations may also compete for behavioural expression. Amino acid starvation, especially glutamine starvation, increases the activity of DA-WED neurons and, remarkably, also drastically lengthens their medial branches in a form of structural plasticity. The medial branches appear to contact the FB-LAL neurons, whose activity drives persistent protein intake. By contrast, the lateral branches of DA-WED neurons form synapses with PLP neurons that drive sugar intake. The dopamine released from DA-WED neurons activates FB-LAL neurons via the DopR2 receptor, but it inhibits PLP neurons via the DopR1

receptor. As yet, how DA-WED neurons sense the need for amino acids has not been elucidated.

Some neurons responsible for feeding behaviour regulate both sugar and amino acid consumption. Six neurons expressing diuretic hormone 44 (DH44) in the adult fly brain regulate consumption of nutritious sugar and essential amino acids (figure 2a) [104,105]. DH44 is a homologue of mammalian corticotrophin-releasing hormone (CRH). These DH44 neurons are located in the PI region. The DH44 neurons are directly activated by nutritious sugars and three specific amino acids: L-glutamate, L-alanine and L-aspartate. Stimulation of downstream neurons expressing DH44 receptor 1 (DH44 R1) leads to rapid proboscis extension, even in the absence of food. Therefore, DH44 neurons have been proposed to function as a post-ingestive nutrient sensor that facilitates the consumption of nutritious sugar and amino acids [104,105]. Neither activation nor silencing of DH44 neurons has an effect on the amount of food ingested, suggesting that they do not mediate general signals of hunger or satiety. Interestingly, although promotion of feeding behaviour by DH44 is independent of internal nutritional status, flies only prefer nutritious sugars over sweet-only sugars when they are starved. Therefore, parts of the DH44 pathway only seem to work in the hungry state. How the DH44 pathway is regulated by starvation is not currently known. However, DH44 neurons express hugin receptor, so they may be directly modulated by the satiety signal hugin [65].

## 4.5. Hunger modulates larval feeding circuits

Hunger also changes the feeding behaviour of fly larvae. Starved larvae have higher feeding rates and increased tolerance for low-quality and noxious foods [15,38,106]. These hunger-driven increases in feeding rate and feeding tolerance are regulated by different neural circuits (figure 2c) [106]. Blocking NPF neurons in starved larvae specifically abolishes increased feeding tolerance without affecting the feeding rate, and the opposite scenario is observed when octopaminergic (OA) neurons are blocked. Blocking the neurotransmission of NPF receptor (NPFR)-expressing neurons also impairs the increased feeding tolerance of starved larvae, whereas overexpression of npfr in NPFR neurons makes larvae feed on low-quality or noxious food even when they are not food-deprived [15,38]. Down- or upregulation of insulin signalling in NPFR neurons also leads to increased or decreased acceptance, respectively, of low-quality or noxious food. Therefore, NPF and DILPs work together on NPFR neurons to regulate feeding tolerance [15,38]. The OA neurons regulate feeding rate through two populations of downstream neurons that express distinct OA receptors, Octβ3R and OAMB [106,107]. Starved larvae with reduced Octβ3R activity cannot increase their feeding rate [106]. By contrast, knockdown of oamb increases feeding rate in larvae fed ad libitum [107]. Therefore, the downstream neurons expressing these two OA receptors contrastingly modulate larval feeding rate. Interestingly, two distinct populations of OA neurons in the larval SEZ that receive gustatory inputs have also been found to contrastingly regulate feeding rate [106]. Laser ablation of five OA-VUM1 neurons caused increased feeding rates in fed larvae, whereas laser ablation of six OA-VUM2 neurons attenuated the elevated feeding rate of starved larvae. Accordingly, the OA-VUM1 and OA-VUM2 neurons are thought to operate upstream of the OAMB- and Octβ3R-expressing neurons, respectively.

Furthermore, apart from regulating feeding tolerance through NPFR neurons, IPCs and DILPs negatively regulate feeding rate in both fed and starved larvae [38]. However, whether they act through the OA neurons remains to be determined.

# 5. Hunger-based modulation of olfactory circuits

Flies mainly rely on olfaction when foraging for food. Hungry flies are more sensitive to and more attracted by food odours [6,32,50]. As for their control of feeding behaviour, hunger and satiety signals modulate both peripheral and central neural circuits to regulate the responses of flies to food odour.

## 5.1. Hunger sensitizes food odour responses

Starvation directly enhances the sensitivity of odorant receptor neurons (ORNs) that detect food odour (figure 3a) [6]. Starved but not fed adult flies exhibit strong seeking behaviour when they smell the food odour emanating from 1% apple cider vinegar (ACV). ACV-responsive ORNs that trigger odour attraction show an enhanced response to ACV upon starvation. Knockdown of sNPF or its receptor (sNPFR) in these ORNs eliminates this enhanced response in hungry flies and the flies exhibit reduced ACV-seeking behaviour. By contrast, overexpression of sNPFR in ACV-responsive ORNs is sufficient to increase their ACV response and evoke ACV-seeking behaviour in fed flies. The sNPF signalling in these ORNs appears to be regulated by satiety-inducing DILPs. ACV-responsive ORNs express InR, activation of which inhibits sNPFR expression. When flies are starved, low levels of DILPs results in high sNPFR expression in the ACV-responsive ORNs, with the increased sNPF signalling in turn enhancing the sensitivity of these ORNs. Therefore, global insulin signalling and the local sNPF pathway work cooperatively in peripheral sensory neurons to tune the sensitivity of flies to food odour.

## 5.2. Hunger desensitizes responses to 'bad' smells

High concentrations of ACV also activate ORNs expressing Or85a that signals negative valence to render them less attractive to adult flies. Just as starvation dampens bitter GRNs in feeding circuits [43], food deprivation also reduces the ACV response in Or85a neurons (figure 3a) [39]. RNA profiling experiments have revealed that transcripts of *Drosophila* tachykinin receptor (DTKR) are increased in the antennae of starved flies. Knockdown of DTKR in the Or85a neurons of starved flies increases the response of those neurons to high concentrations of ACV. The same manipulation also reduces the odour-seeking response of hungry flies. The source of the DTKR ligand tachykinin (TK) is local interneurons (LNs) in the antennal lobe, where the axons of Or85a neurons also project to. TK knockdown in the LNs results in the same phenotype as DTKR knockdown in Or85a neurons. Furthermore, enhancing insulin signalling in Or85a neurons increases their activity and decreases the attraction of hungry flies to a high-concentration ACV. Although the direct link between insulin signalling and the expression of DTKR has not been established, it may be that reduced insulin signalling in starved flies leads to increased expression of DTKR, which suppresses the response of Or85a neurons to high concentrations of ACV upon receiving

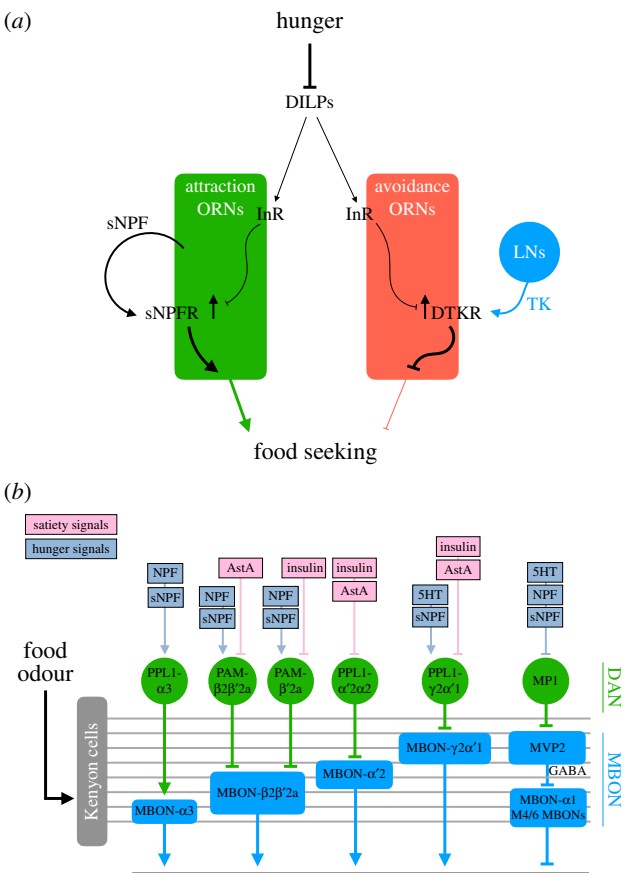

**Figure 3.** Hunger-based control of olfactory circuits. (*a*) Starvation inhibits the release of DILPs, reducing insulin signalling in both attraction and avoidance ORNs. The reduced insulin signalling leads to increased expression of sNPFR and DTKR in the attraction and avoidance ORNs, respectively. In the attraction ORNs, sNPFR receives sNPF secreted from the same ORNs, which in turn enhances their attraction to food odours. By contrast, in avoidance ORNs, DTKR receives TK from LNs in the antennal lobe and consequently repress the synaptic output of avoidance ORNs. (*b*) Five MBON pathways (blue) that innervate different zones of the KC axons promote odour-driven food-seeking behaviour. These MBON pathways are regulated by their corresponding dopaminergic neurons (DANs; green), and these DANs receive different combinations of hunger and satiety signals. The MP1-MVP2-M4/6 MBON pathway also mediates hunger control of learned food-seeking behaviour.

TK released from the LNs. Starvation therefore fine-tunes the sensitivity of flies to food odour through bi-directional regulation of ORNs with opposite valences.

## 5.3. Hunger modulates responses to learned food-associated olfactory cues

In addition to modulating the innate responses of flies to food odour, satiety state also gates their expression of sugar-rewarded olfactory memory [50,108]. As we have already mentioned, starved adult flies can be trained to form an appetitive olfactory memory by pairing odours with sugar reward. After training, flies only approach the sugar-predictive odours when they are hungry. Starvation promotes this learned approach behaviour via the hunger signal NPF. NPF executes its function by inhibiting two pairs of dopaminergic neurons (MP1) innervating the mushroom body (MB), an olfactory memory centre in the fly brain [50]. The adult MB has five

royalsocietypublishing.org/journal/rsob    Open Biol. 9: 180259

lobes, which are axonal bundles of approximately 2500 intrinsic neurons called Kenyon cells (KCs). The MB lobes can be subdivided into 15 zones innervated by 20 types of dopaminergic neurons and 21 types of output neurons (MBONs). The axons of different dopaminergic neuronal types and the dendrites of different MBONs innervate distinct zones of the MB lobes [109]. The KCs receive direct input from olfactory projection neurons, representing the third-order neurons in the olfactory pathway. Different MBONs receive odour information from different zones along the MB lobes, and the dopaminergic neurons are thought to modify the KC-MBON synapses in zone-specific ways [109,110]. Some MBONs encode positive or negative valences, and olfactory learning is thought to tip over the balanced collective outputs of these positives and negatives, resulting in odour-driven approach or avoidance behaviour [110,111]. Knockdown of NPF in MP1 neurons or artificially activating MP1 neurons impairs the learned approach to the sugar-predictive odours in hungry flies, whereas blocking MP1 neuronal output promotes the learned approach in fed flies. Therefore, MP1 neuronal activity blocks learned food-seeking behaviour, and NPF can remove this blockage by downregulating MP1 neuronal activity. The neurons downstream of MP1 are GABAergic MBONs called MVP2 neurons. Dopamine released from MP1 neurons depresses the odour-evoked responses of MVP2 neurons. The axons of MVP2 are projected to the tips of the MB horizontal lobes where they release GABA to inhibit another group of negative-valence MBONs called the M4/6 neurons [112]. Thus, MP1-MVP2-M4/6 neurons represent a multi-inhibitory neural circuit that represses odour-driven learned food-seeking behaviour in fed flies. When MP1 neuronal activity is downregulated by NPF upon starvation, high MVP2 activity inhibits M4/6 neurons to permit learned food-seeking behaviour (figure 3b).

## 5.4. Hunger modulates MB circuits to tune innate responses to food odour

The MB not only controls the responses of adult flies to learned odours, it also regulates their innate food-seeking behaviour evoked by food odour (figure 3b). A recent study shows that 5 of the 21 types of MBONs are required for hungry flies to seek food odours [32]. The MP1-MVP2-M4/6 pathway has also been identified as regulating innate food-seeking behaviour. Moreover, MP1 neurons are not only regulated by NPF but also by two other hunger signals: sNPF and serotonin. Knockdown of NPFR, sNPFR or the serotonin receptor 5H2A in MP1 neurons impairs innate food-seeking behaviour. Four other MBONs—MBON-α3, MBON-β2β′2a, MBON-α′2 and MBON-γ2α′1—and their corresponding dopaminergic neurons also regulate innate food-seeking behaviour. Blocking these MBONs and dopaminergic neurons diminishes innate food-seeking behaviour in hungry flies, and activation of the dopaminergic neurons is sufficient to evoke food-seeking behaviour in fed flies. Importantly, results from RNAi knockdown of various receptors for different hunger and satiety signals support the idea that the MB-innervating dopaminergic neurons are directly regulated by many of these signals, making the MB an integrative centre for hunger and satiety signals in the fly brain. This idea is further supported by a recent study showing that the MB circuit also regulates fat storage and food intake [113].

## 6. Hunger-based modulation of locomotion circuits

Starvation increases locomotion, presumably to facilitate food-seeking behaviour [4,89]. This starvation-induced hyperactivity in adult flies requires the AKH hunger signal [4,33]. Flies lacking AKH or its receptor AKHR do not exhibit starvation-induced hyperactivity. AKH stimulates locomotion by acting on a small number (two to four neurons per hemisphere) of AKHR-expressing neurons in the SEZ [33]. Silencing these AKHR neurons abolishes the hyperactivity induced by starvation, whereas activating them accelerates the onset of hyperactivity upon starvation. These AKHR neurons are specific for controlling locomotion since manipulating them has no effect on feeding behaviour. These AKHR neurons also express InR and their activity is suppressed by DILPs. Thus, like ISNs that regulate feeding behaviour, these AKHR neurons are directly influenced by both hunger and satiety signals [16,33]. These AKHR neurons are octopaminergic, but their downstream neurons await identification.

## 7. Intricacy of foraging and feeding behaviour: a future challenge

Recently, detailed quantitative methods measuring proboscis extensions to food (sips) and consumed food volume in close to real time in adult flies have revealed the intricate microstructure of feeding modulated by starvation [8,102,114]. Adult fly feeding is highly rhythmic, with most sips lasting for 0.16 s, with an inter-sip interval (ISI) of 0.08 s when consuming 10% sucrose [114]. The sips can be further organized into bursts, each defined as three or more consecutive sips separated by inter-burst intervals (IBI) shorter than double the median ISI. Although flies increase food consumption when starved, the duration of sips and the ISI remain nearly constant. However, starved flies exhibit higher numbers of sips per burst and a shorter IBI. Interestingly, the dynamics of sips per burst and IBI are differentially modulated by hunger. Four hours of starvation significantly shortens IBI without affecting the number of sips. An additional four hours of starvation (8 h in total) does not further curtail IBI, but it significantly increases the number of sips per burst. Furthermore, 4 h starvation is sufficient to lengthen the time flies spend on a food patch during each visit (activity bout duration), but 8 h starvation is needed to increase the length of the feed bursts. When 8 h-starved flies eat to satiation, increased IBI and decreased activity bout duration occur within 10 min after the meal starts, whereas a decrease in number of sips per burst can only be detected near the end of the meal. Similar regulation of the dynamics of feed microstructure has also been observed in protein-starved flies eating yeast [8]. Furthermore, detailed analysis of adult female flies foraging on distributed yeast patches uncovered modulation of foraging decisions by internal amino acid-deprivation state [8]. When amino acid demand is low (e.g. in virgin females), amino acid-deprived flies spend more time on a yeast patch during each yeast visit, but their rate of yeast encounters and probability of stopping at a yeast patch are not different from amino acid-satiated flies. All three of these parameters are increased when mated female flies (experiencing high amino acid demand) are deprived of amino acids. Amino acid deprivation also

modulates the exploratory and exploitatory behaviour of flies. Mated female flies pretreated with amino acid-rich food typically explore large area of yeast patches and almost never return to yeast patches they have just visited. By contrast, amino acid-deprived mated female flies show reduced global exploration and have a much higher likelihood of revisiting the same yeast patch [8]. This local food searching behaviour appears to rely on idiothetic cues independent of vision and olfaction [115]. These detailed analyses of feeding and foraging behaviour have revealed the sophisticated nature of hunger in tuning rich repertoires of behaviour modules. An important future challenge is to understand mechanistically how this level of control is achieved.

## 8. Concluding remarks

Studying hunger and the behaviours it elicits will not only lead to a better understanding of this highly conserved primary motivation, but will also provide insights into many general and fundamental questions in neuroscience, such as how the nervous system senses and encodes bodily requirements, how the brain integrates external stimuli and internal states, how a neural circuit gates information flow, and how goal-directed behaviour emerges from the brain. Studies of hunger-driven behaviour in the fruit fly have helped to reveal the biological nature of hunger at the molecular and neural circuit levels. Several important principles can be deduced from these studies. First, the brain is equipped with sensors that detect specific nutrients, and these centralized sensors and those in peripheral tissues work in concert to assess bodily requirements. Second, hunger and satiety states are encoded by a large number of neuromodulators, many of which are highly conserved across animal species. These neuromodulators interact to form an intricate regulatory network that presumably gives rise to coherent feeding and foraging behaviour. Third, these hunger and satiety neuromodulators can temporarily and reversibly reconfigure both central and peripheral neural circuits so that the same inputs can lead to different outputs in a state-dependent manner. However, our quest for a complete understanding of hunger-driven behaviour has only just begun. Sophisticated behavioural tracking/ analysis techniques have revealed intricate behavioural microstructures modulated by hunger. What are the neural-circuit underpinnings of these behavioural microstructures? How are they influenced by hunger and satiety neuromodulators? And, most importantly, how are these microstructures and their control circuits integrated to give rise to coordinated feeding and foraging behaviour? These are important and very challenging questions. Nevertheless, recent technological breakthroughs—including whole-brain functional imaging of live flies, advanced behavioural tracking, improved genetically encoded sensors of neurotransmitter release and neuromodulator activity, and real-time optogenetic control of defined neuronal types—have brought the answers to these questions closer to our grasp.

Data accessibility. This article has no additional data.
Competing interests. We declare we have no competing interests.
Funding. This study was supported by a grant from Taiwan Ministry of Science and Technology (105-2628-B-001-005-MY3) to S.L..

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
