## [Reviewer comments · Open Biology]

Review History

RSOB-18-0259.R0 (Original submission)

Review form: Reviewer 1

Recommendation

Accept with minor revision (please list in comments)

Are each of the following suitable for general readers?

- a) **Title**
Yes
- b) **Summary**
Yes
- c) **Introduction**
Yes

Is the length of the paper justified?

Yes

Should the paper be seen by a specialist statistical reviewer?

No

Is it clear how to make all supporting data available?

Not Applicable

Is the supplementary material necessary; and if so is it adequate and clear?

Not Applicable

Do you have any ethical concerns with this paper?

No

Comments to the Author

In this manuscript, Lin and colleagues present a thorough review of the state-of-the-art in the research of how neural circuits control *Drosophila* feeding behavior based on their hunger state. The strength of the review is the provided extensive overview of the current knowledge on (1) how nutritional needs are sensed by different tissues including the brain; (2) how nutrient status is communicated to the brain; (3) the neuromodulatory and peptidergic pathways involved in encoding of the hunger state and; (4) how the activity of feeding circuits (from sensory neurons to locomotor circuits) are regulated by different modulatory pathways based on the nutritional state of the fly. They have nicely discussed our understanding of how the intake of different macronutrients (i.e. sugar and protein) could be controlled by modulating the activity of different sensory circuits involved in different aspects of the feeding program (e.g. modulation of olfaction and locomotion for foraging, gustatory sensory neurons for food exploitation). They have also mentioned higher order feeding circuits such as mushroom body and central complex. The writing and figures are clear.

There are four major points which I would like to raise and which in my opinion could help the authors to significantly improve the manuscript:

At this stage the comprehensiveness of the discussed players controlling feeding makes the different parts of the review a bit difficult to read. Sometimes one feels like crossing a long desert of facts. I think that subdividing the review in subsections would help. For example the part on the feeding circuits could be subdivided into taste and non-taste parts. I also don't think that the memory part should necessarily be lumped together with olfaction. A clearer structure would help the reader to orient themselves in the sea of facts.

A major disappointment is that while the title, abstract and the beginning of the review emphasizes behavior, there is no further emphasis or discussion on the different behavioral elements underlying foraging and feeding. The behavioral manifestation of the hunger state in terms of the changes in the foraging and feeding behavior is not well discussed. A section on how deprivation of nutrients affects foraging (e.g. Corrales-Carvajal et al., 2016, Kim and Dickinson 2017) and the feeding microstructure (e.g. Itskov et al., 2014, Yapici et al., 2016) would be useful to clarify the behavioral relevance of the neuromodulation etc.. This could also give the non-expert reader a feeling of the behavioral approaches and assays used to mechanistically dissect feeding and could highlight that a key challenge for the future will be to assign how the described neuromodulators/neurons control foraging and feeding.

In general I also would suggest that the authors emphasize that what is known in larvae and adults does not necessarily have to be the same. This is a common misconception in papers and reviewers. I appreciated that the authors tried to always mention if data were generated in larvae or adults but I think this could be emphasized even more. Maybe they could even emphasize that in a specific paragraph. Also in figure 1 which I think at this stage lumps together data from larvae and adults the authors should be careful to separate if the conclusion comes from larvae or adults. If this is the case, maybe splitting the diagram into two like done in figure 2 would be better.

Finally, at the beginning of the review (for example the introduction) the authors provide a lot of general concepts and ideas about feeding, foraging, and how it is regulated. Unfortunately no citations are provided for these generally accepted "facts". Unfortunately, these aspects are foundational for how we think about feeding and foraging and are unfortunately not always well documented if at all. The authors should therefore provide citations for these initial statements.

MINOR POINTS:

Although the modulatory pathways controlling the hunger-driven behaviors are well-defined, the neural circuits that are involved in the sensorimotor transformation are not well-discussed (e.g. interneurons and PERin neurons previously defined by the Kristin Scott lab, Fdg command neuron identified by Flood et al., 2013). Besides being at the core of the sensorimotor transformations controlling foraging and feeding behaviors, these neurons are likely targets of neuromodulation as well. It would be useful if the authors could add a paragraph discussing these neural elements and their potential interactions with the modulatory circuits.

As a general remark I am not sure that it is always correct to call some of the molecules "homologues" of vertebrate molecules. I think in some cases "orthologues" would be a better choice.

poxn mutants are not taste blind as they still have internal taste organs and also taste pegs are not affected by the mutation. In general the field has become more careful in interpreting the results from "taste mutants" as an evidence for postingestive nutrient assessment.

As for the concluding remarks, it would be essential to speculate how important it is to study the integrated action of different modulatory pathways control different aspects of the fly feeding program ranging from foraging dynamics to the microstructure of food consumption.

Decision letter (RSOB-18-0259.R0)

16-Jan-2019

Dear Dr Lin,

We are pleased to inform you that your manuscript RSOB-18-0259 entitled "Neural basis of hunger-driven behavior in *Drosophila*" has been accepted by the Editor for publication in Open Biology. The reviewer(s) have recommended publication, but also suggest some minor revisions to your manuscript. Therefore, we invite you to respond to the reviewer(s)' comments and revise your manuscript.

Please submit the revised version of your manuscript within 14 days. If you do not think you will be able to meet this date please let us know immediately and we can extend this deadline for you.

When submitting your revised manuscript, you will be able to respond to the comments made by the referee(s) and upload a file "Response to Referees" in "Section 6 - File Upload". You can use this to document any changes you make to the original manuscript. In order to expedite the

processing of the revised manuscript, please be as specific as possible in your response to the referee(s).

- 1) A text file of the manuscript (doc, txt, rtf or tex), including the references, tables (including captions) and figure captions. Please remove any tracked changes from the text before submission. PDF files are not an accepted format for the "Main Document".
- 2) A separate electronic file of each figure (tiff, EPS or print-quality PDF preferred). The format should be produced directly from original creation package, or original software format. Please note that PowerPoint files are not accepted.
- 3) Electronic supplementary material: this should be contained in a separate file from the main text and meet our ESM criteria (see <http://royalsocietypublishing.org/instructions-authors#question5>). All supplementary materials accompanying an accepted article will be treated as in their final form. They will be published alongside the paper on the journal website and posted on the online figshare repository. Files on figshare will be made available approximately one week before the accompanying article so that the supplementary material can be attributed a unique DOI.

Online supplementary material will also carry the title and description provided during submission, so please ensure these are accurate and informative. Note that the Royal Society will not edit or typeset supplementary material and it will be hosted as provided. Please ensure that the supplementary material includes the paper details (authors, title, journal name, article DOI). Your article DOI will be 10.1098/rsob.2016[[last 4 digits of e.g. 10.1098/rsob.20160049].

- 4) A media summary: a short non-technical summary (up to 100 words) of the key findings/importance of your manuscript. Please try to write in simple English, avoid jargon, explain the importance of the topic, outline the main implications and describe why this topic is newsworthy.

Images

Data-Sharing

It is a condition of publication that data supporting your paper are made available. Data should be made available either in the electronic supplementary material or through an appropriate repository. Details of how to access data should be included in your paper. Please see <http://royalsocietypublishing.org/site/authors/policy.xhtml#question6> for more details.

Sincerely,

The Open Biology Team
<mailto:openbiology@royalsociety.org>

Reviewer(s)' Comments to Author:

Referee: 1

Comments to the Author(s)

In this manuscript, Lin and colleagues present a thorough review of the state-of-the-art in the research of how neural circuits control *Drosophila* feeding behavior based on their hunger state. The strength of the review is the provided extensive overview of the current knowledge on (1) how nutritional needs are sensed by different tissues including the brain; (2) how nutrient status is communicated to the brain; (3) the neuromodulatory and peptidergic pathways involved in encoding of the hunger state and; (4) how the activity of feeding circuits (from sensory neurons to locomotor circuits) are regulated by different modulatory pathways based on the nutritional state of the fly. They have nicely discussed our understanding of how the intake of different macronutrients (i.e. sugar and protein) could be controlled by modulating the activity of different sensory circuits involved in different aspects of the feeding program (e.g. modulation of olfaction and locomotion for foraging, gustatory sensory neurons for food exploitation). They have also mentioned higher order feeding circuits such as mushroom body and central complex. The writing and figures are clear.

There are four major points which I would like to raise and which in my opinion could help the authors to significantly improve the manuscript:

At this stage the comprehensiveness of the discussed players controlling feeding makes the different parts of the review a bit difficult to read. Sometimes one feels like crossing a long desert of facts. I think that subdividing the review in subsections would help. For example the part on the feeding circuits could be subdivided into taste and non-taste parts. I also don't think that the memory part should necessarily be lumped together with olfaction. A clearer structure would help the reader to orient themselves in the sea of facts.

A major disappointment is that while the title, abstract and the beginning of the review emphasizes behavior, there is no further emphasis or discussion on the different behavioral elements underlying foraging and feeding. The behavioral manifestation of the hunger state in terms of the changes in the foraging and feeding behavior is not well discussed. A section on how deprivation of nutrients affects foraging (e.g. Corrales-Carvajal et al., 2016, Kim and Dickinson 2017) and the feeding microstructure (e.g. Itskov et al., 2014, Yapici et al., 2016) would be useful to clarify the behavioral relevance of the neuromodulation etc.. This could also give the non-expert reader a feeling of the behavioral approaches and assays used to mechanistically dissect feeding and could highlight that a key challenge for the future will be to assign how the described neuromodulators/neurons control foraging and feeding.

In general I also would suggest that the authors emphasize that what is known in larvae and adults does not necessarily have to be the same. This is a common misconception in papers and reviewers. I appreciated that the authors tried to always mention if data were generated in larvae or adults but I think this could be emphasized even more. Maybe they could even emphasize that in a specific paragraph. Also in figure 1 which I think at this stage lumps together data from larvae and adults the authors should be careful to separate if the conclusion comes from larvae or adults. If this is the case, maybe splitting the diagram into two like done in figure 2 would be better.

Finally, at the beginning of the review (for example the introduction) the authors provide a lot of general concepts and ideas about feeding, foraging, and how it is regulated. Unfortunately no citations are provided for these generally accepted "facts". Unfortunately, these aspects are

foundational for how we think about feeding and foraging and are unfortunately not always well documented if at all. The authors should therefore provide citations for these initial statements.

MINOR POINTS:

Although the modulatory pathways controlling the hunger-driven behaviors are well-defined, the neural circuits that are involved in the sensorimotor transformation are not well-discussed (e.g. interneurons and PERin neurons previously defined by the Kristin Scott lab, Fdg command neuron identified by Flood et al., 2013). Besides being at the core of the sensorimotor transformations controlling foraging and feeding behaviors, these neurons are likely targets of neuromodulation as well. It would be useful if the authors could add a paragraph discussing these neural elements and their potential interactions with the modulatory circuits.

As a general remark I am not sure that it is always correct to call some of the molecules "homologues" of vertebrate molecules. I think in some cases "orthologues" would be a better choice.

poxn mutants are not taste blind as they still have internal taste organs and also taste pegs are not affected by the mutation. In general the field has become more careful in interpreting the results from "taste mutants" as an evidence for postingestive nutrient assessment.

As for the concluding remarks, it would be essential to speculate how important it is to study the integrated action of different modulatory pathways control different aspects of the fly feeding program ranging from foraging dynamics to the microstructure of food consumption.

Author's Response to Decision Letter for (RSOB-18-0259.R0)

See Appendix A.

Decision letter (RSOB-18-0259.R1)

04-Mar-2019

Dear Dr Lin

We are pleased to inform you that your manuscript entitled "Neural basis of hunger-driven behavior in *Drosophila*" has been accepted by the Editor for publication in Open Biology.

Sincerely,

The Open Biology Team
mailto: openbiology@royalsociety.org

Appendix A

Point-by-point response to reviewers

We thank the reviewers for their helpful comments and suggestions to improve the manuscript. We have followed their suggestions closely and revised the manuscript accordingly. We believe these changes have significantly improved the manuscript. For ease of reading, our responses to comments are provided in blue font.

There are four major points which I would like to raise and which in my opinion could help the authors to significantly improve the manuscript:

At this stage the comprehensiveness of the discussed players controlling feeding makes the different parts of the review a bit difficult to read. Sometimes one feels like crossing a long desert of facts. I think that subdividing the review in subsections would help. For example, the part on the feeding circuits could be subdivided into taste and non-taste parts. I also don't think that the memory part should necessarily be lumped together with olfaction. A clearer structure would help the reader to orient themselves in the sea of facts.

We thank the reviewer for this helpful comment. We have improved the structure of the manuscript by subdividing into subsections some parts of the review.

A major disappointment is that while the title, abstract and the beginning of the review emphasizes behavior, there is no further emphasis or discussion on the different behavioral elements underlying foraging and feeding. The behavioral manifestation of the hunger state in terms of the changes in the foraging and feeding behavior is not well discussed. A section on how deprivation of nutrients affects foraging (e.g. Corrales-Carvajal et al., 2016, Kim and Dickinson 2017) and the feeding microstructure (e.g. Itskov et al., 2014, Yapici et al., 2016) would be useful to clarify the behavioral relevance of the neuromodulation etc. This could also give the non-expert reader a feeling of the behavioral approaches and assays used to mechanistically dissect feeding and could highlight that a key challenge for the future will be to assign how the described neuromodulators/neurons control foraging and feeding.

We thank the reviewer for these valuable suggestions. We have added a new section, "Intricacy of foraging and feeding behavior: a future challenge," to cover these points.

In general I also would suggest that the authors emphasize that what is known in larvae and adults does not necessarily have to be the same. This is a common misconception in papers and reviewers. I appreciated that the authors tried to always mention if data were generated in larvae or adults but I think this could be emphasized even more. Maybe they could even emphasize that in a specific paragraph. Also in figure 1 which I think at this stage lumps together data from larvae and adults the authors should be careful to separate if the conclusion comes from larvae or adults. If this is the case, maybe splitting the diagram into two like done in figure 2 would be better.

We agree with the reviewer that it is important to distinguish whether a conclusion is based on data from larvae or adults. However, we do think that there are many similarities between larvae and adults in terms of neuromodulator functions. Accordingly, we still prefer to discuss larval and adult findings together. Nevertheless, we have revised our introduction to emphasize the potential differences in neural mechanisms between larvae and adult flies. We have further ensured that we always categorically state if a data is from larvae or adults. We have also now labeled the citations in Fig. 1 differently to distinguish larval and adult studies.

Finally, at the beginning of the review (for example the introduction) the authors provide a lot of

general concepts and ideas about feeding, foraging, and how it is regulated. Unfortunately no citations are provided for these generally accepted “facts”. Unfortunately, these aspects are foundational for how we think about feeding and foraging and are unfortunately not always well documented if at all. The authors should therefore provide citations for these initial statements.

We thank the reviewer for this helpful suggestion. We have now provided citations for these statements.

MINOR POINTS:

Although the modulatory pathways controlling the hunger-driven behaviors are well-defined, the neural circuits that are involved in the sensorimotor transformation are not well-discussed (e.g. interneurons and PERin neurons previously defined by the Kristin Scott lab, Fdg command neuron identified by Flood et al., 2013). Besides being at the core of the sensorimotor transformations controlling foraging and feeding behaviors, these neurons are likely targets of neuromodulation as well. It would be useful if the authors could add a paragraph discussing these neural elements and their potential interactions with the modulatory circuits.

We have added a new subsection to cover these findings.

As a general remark I am not sure that it is always correct to call some of the molecules “homologues” of vertebrate molecules. I think in some cases “orthologues” would be a better choice.

We revised our manuscript to remove “homolog” where we deem it unsuitable. However, in most cases where we have confirmed our use of this term is consistent with the literature, we have maintained it. To our knowledge, ortholog can be included in the definition of homolog (i.e. homologs can be subdivided into orthologs and paralogs).

poxn mutants are not taste blind as they still have internal taste organs and also taste pegs are not affected by the mutation. In general the field has become more careful in interpreting the results from “taste mutants” as an evidence for postingestive nutrient assessment.

We thank the reviewer for this helpful suggestion. We have removed the statement about poxn mutants.

As for the concluding remarks, it would be essential to speculate how important it is to study the integrated action of different modulatory pathways control different aspects of the fly feeding program ranging from foraging dynamics to the microstructure of food consumption.

We again thank the reviewer for this valuable suggestion and have revised our concluding remarks accordingly.